# Immunosuppression via adenosine receptor activation by adenosine monophosphate released from apoptotic cells

Hiroshi Yamaguchi[1], Toshihiko Maruyama[2], Yoshihiro Urade[2†], Shigekazu Nagata[1,3]*

[1]Department of Medical Chemistry, Graduate School of Medicine, Kyoto University, Kyoto, Japan; [2]Department of Molecular Behavioral Biology, Osaka Bioscience Institute, Osaka, Japan; [3]Core Research for Evolutional Science and Technology, Japan Science and Technology Corporation, Kyoto, Japan

*For correspondence: snagata@mfour.med.kyoto-u.ac.jp

Present address: †Molecular Sleep Biology Laboratory, WPI-International Institute for Integrative Sleep Medicine, University of Tsukuba, Ibaraki, Japan

Competing interests: The authors declare that no competing interests exist.

**Abstract** Apoptosis is coupled with recruitment of macrophages for engulfment of dead cells, and with compensatory proliferation of neighboring cells. Yet, this death process is silent, and it does not cause inflammation. The molecular mechanisms underlying anti-inflammatory nature of the apoptotic process remains poorly understood. In this study, we found that the culture supernatant of apoptotic cells activated the macrophages to express anti-inflammatory genes such as *Nr4a* and *Thbs1*. A high level of AMP accumulated in the apoptotic cell supernatant in a Pannexin1-dependent manner. A nucleotidase inhibitor and A2a adenosine receptor antagonist inhibited the apoptotic supernatant-induced gene expression, suggesting AMP was metabolized to adenosine by an ecto-5'-nucleotidase expressed on macrophages, to activate the macrophage A2a adenosine receptor. Intraperitoneal injection of zymosan into *Adora2a-* or *Panx1*-deficient mice produced high, sustained levels of inflammatory mediators in the peritoneal lavage. These results indicated that AMP from apoptotic cells suppresses inflammation as a 'calm down' signal.

## Introduction

Approximately, 100,000 cells undergo apoptosis every second in our bodies, and are engulfed by macrophages and immature dendritic cells (*Fuchs and Steller, 2011*). Inefficient clearance of apoptotic cells causes systemic lupus erythematosus (SLE)-type autoimmune disease in humans and mice (*Botto et al., 1998*; *Hanayama et al., 2004*; *Munoz et al., 2010*). Apoptotic cells characteristically expose 'eat me' signal(s) to macrophages (*Nagata et al., 2010*; *Ravichandran, 2011*), and one of the primary 'eat me' signals is phosphatidylserine (PtdSer). PtdSer is a phospholipid that localizes to the inner leaflet of plasma membranes in healthy cells; however, when cells undergo apoptosis, it is exposed on the outer cell surface in a caspase-dependent manner (*Fadok et al., 2001*; *Leventis and Grinstein, 2010*; *Suzuki et al., 2013*). Macrophages express membrane proteins that function as receptors for PtdSer (e.g., Tim-4, stabllin 2, and BAI1), or opsonins that bind to PtdSer (e.g. MFG-E8, Protein S, and Gas6) (*Nagata et al., 2010*). Masking the PtdSer on apoptotic cells prevents their engulfment by macrophages, supporting the idea that PtdSer is an important 'eat me' signal.

Apoptotic cells also produce 'find me' signals to attract macrophages (*Ravichandran, 2011*). Lysophosphatidylcholine (LPC) (*Lauber et al., 2003*), ATP/UTP (*Elliott et al., 2009*), fractalkine (*Truman et al., 2008*), and sphingosine-1-phosphate (S1P) (*Gude et al., 2008*) are released from apoptotic cells in a caspase-dependent manner. Macrophages express specific receptors for these signals (P2Y$_2$ for ATP and UTP, CX3CR1 for fractalkine, and S1PR1 for S1P), which may mediate migration to the dying

**eLife digest** Infections, toxins, and trauma can all injure tissue and cause the cells inside the tissue to die. When a cell dies, the membrane that surrounds it ruptures and its contents spill out, triggering inflammation of the surrounding tissues. This inflammation is part of the body's efforts to begin the healing process but, if left uncontrolled, inflammation itself can cause further tissue damage.

Diseased or damaged cells can also 'choose' to kill themselves to protect other healthy cells. This process, which is called apoptosis, also eliminates about 100,000 cells that are too old, or just no longer needed, from the human body every second. A cell undergoing apoptosis essentially dismantles itself, and the remains of the cell are packaged up, and cleared away by the white blood cells. Interestingly, this programed cell death releases many of the same molecules as other dying cells, but does so without triggering inflammation. The reason behind this lack of inflammation has not been clear.

Now, Yamaguchi et al. have addressed this issue, and shown that cells undergoing apoptosis also release a chemical called adenosine monophosphate (AMP) that acts as a 'calm down' signal. The AMP is processed by white blood cells to a simpler chemical, which 'switches on' various genes in the white blood cells. This leads to the production of proteins that suppress the inflammation that would otherwise be triggered by other molecules released from the cells undergoing apoptosis. The findings of Yamaguchi et al. show how the community of cells in our body is kept in a healthy balance, and in the future, could improve our understanding and the treatment of inflammatory diseases.

cells (*Ravichandran, 2011*). The 'find me' signals are thought to prime macrophages for engulfment, for example by enhancing the expression of MFG-E8 (*Miksa et al., 2007*). On the other hand, LPC, ATP/UTP, and S1P are known to cause inflammation (*Trautmann, 2009*; *Maceyka et al., 2012*; *Meyer zu Heringdorf and Jakobs, 2007*), which may contradict the anti-inflammatory nature of the apoptotic process. In addition, several groups showed that apoptotic cells produce growth-stimulating molecules such as Wnt3 or prostaglandin E2 (*Chera et al., 2009*; *Huang et al., 2011*; *Mollereau et al., 2013*), which is also against the silent nature of the process. Thus, how the anti-inflammatory character of the apoptosis is maintained during the process has been unclear.

In this study, we found that soluble factors derived from the apoptotic cells activated macrophages to express immediate early response genes including anti-inflammatory genes such as *Nuclear receptor 4A* (*Nr4a*) and *Thrombospondin* (*Thbs*)*1*. Using *Thbs1* gene expression as a marker, the molecule responsible for the gene expression was identified as AMP. The overexpression of Pannexin 1 in apoptotic cells accelerated the accumulation of AMP in the culture supernatant of apoptotic cells, while a caspase-resistant form of Pannexin 1 inhibited the AMP accumulation. An inhibitor of ecto-5'-nucleotidase blocked the AMP-induced activation of the *Thbs1* gene in macrophages, while macrophages from A2a adenosine receptor (*Adora2a*)-deficient mice did not respond to AMP. These results indicated that AMP from apoptotic cells was converted to adenosine by an ecto-5'-nucleotidase on macrophages, and that the adenosine activated a set of genes in macrophages via the A2a adenosine receptor. Mice deficient in *Adora2a* or *Panx1* exhibited prolonged peritonitis following intraperitoneal zymosan injection, suggesting that the AMP released from apoptotic peritoneal cells exerted an anti-inflammatory effect by activating the A2a adenosine receptor.

## Results

### Gene expression in macrophages by a factor released from apoptotic cells

If apoptotic cells produce 'danger' or 'anti-danger' signal(s), we rationalized that such signals would activate gene expression in macrophages. To investigate this possibility, we examined the effect of the culture supernatant from apoptotic cells on macrophage gene expression. Mouse WR19L transformants expressing Fas (W3 cells) were treated with Fas ligand (FasL) for 30 min, washed, and then further incubated for 60 min. Following FasL treatment, more than 90% of the W3 cells were

Annexin V positive, and only small percentage were positive for both Annexin V and propidium iodide (PI) (*Figure 1—figure supplement 1*), indicating that the majority of cells had undergone apoptosis but not necrosis. Mouse bone marrow-derived macrophages (BMDMs) were then incubated for 1 hr with the supernatant of FasL-treated W3 cells, and subjected to microarray analysis. As shown in *Figure 1A*, the mRNA levels of *N-myc (Mycn)*, *Nr4a* orphan nuclear receptor family members, *Egr* transcription factors (*Egr2* and *Egr3*), *Thrombospondin 1* (*Thbs1*), and *Il-1β* were 15- to 200-fold higher in the macrophages treated with apoptotic cell supernatant than in the control, untreated macrophages. A real-time RT-PCR analysis confirmed that the supernatants of apoptotic cells but not of healthy cells strongly induced the expression of *Nr4a1*, *Nr4a2*, and *Thbs1* (*Figure 1B*). When W3 cells were treated with FasL in the presence of Q-VD-OPh, a caspase inhibitor (*Caserta et al., 2003*), the ability of the supernatant to upregulate the *Thbs1* gene was abrogated, indicating that the factor(s) responsible for upregulating *Thbs1* gene were generated in a caspase-dependent manner (*Figure 1C*). Thbs1 and Nr4a are known to suppress inflammation (*Lopez-Dee et al., 2011*; *McMorrow and Murphy, 2011*), and a danger signal such as ATP is unlikely to activate these genes.

Treatment of the apoptotic cell supernatant with proteinase K (50 µg/ml for 60 min), DNase I (6 U/ml for 60 min), or RNase A (5 µg/ml for 60 min) did not prevent its ability to enhance *Thbs1* gene expression (*Figure 1D*), suggesting that the factor(s) were not proteins or polynucleotides. When the supernatant was subjected to centrifugal ultrafiltration with a filter with a nominal cutoff of 10 kDa, most of the activity was found in the filtrate, and not in the concentrate (*Figure 1E*). These results indicated that the molecular weight of the factor(s) that activated the macrophages were less than 10 kDa, and may have been present as a free form.

## Identification of AMP as a factor that stimulates gene expression in macrophages

To identify the molecule(s) present in the apoptotic cell supernatant, the supernatant was subjected to LC-MS analysis. As shown in *Figure 2A*, various nucleotides and nucleosides such as AMP, UMP, cytidine, inosine, hypoxanthine, and uridine were present in the FasL-treated cell supernatant.

To examine which molecule(s) in the apoptotic cell supernatant were responsible for the enhanced *Thbs1* gene expression, macrophages were incubated with the various compounds found in the apoptotic cell supernatant at 10 µM. The results indicated that AMP, but none of other nucleosides or nucleotides, activated macrophages (*Figure 2B*). Apyrase, an enzyme that catalyzes the hydrolysis of ATP to AMP, had no effect on the apoptotic cell supernatant activity, indicating that the factor was not ATP or ADP (*Figure 2C*). In contrast, adenosine-5′-O-(α, β- methylenediphosphate) (AOPCP), a non-hydrolyzable ADP analog that inhibits the ecto-5′-nucleotidase, blocked the *Thbs1* gene expression induced by the apoptotic cell supernatant (*Figure 2C*), indicating that adenosine, but not AMP, is responsible for activation of the *Thbs1* gene in macrophages. Treatment of the apoptotic cell supernatant with adenosine deaminase (ADA), that converts adenosine to inosine abrogated its ability to activate *Thbs1* gene expression (*Figure 2C*), confirming that adenosine was responsible for the activity. Accordingly, treating BMDMs with adenosine activated the *Thbs1* gene in a dose-dependent manner (*Figure 2D*). Since adenosine was undetectable in the supernatant of the FasL-treated W3 cells, we concluded that AMP in the supernatant of these cells was converted to adenosine by the ecto-5′-nucleotidase expressed on macrophages, resulting in the stimulation of gene expression in macrophages.

Extracellular adenosine is recognized by the G-protein coupled adenosine receptor family (*Fredholm, 2007*). A real-time RT-PCR analysis indicated that BMDMs and thioglycollate-elicited peritoneal macrophages (thio-pMacs) did not express A1 or A3 adenosine receptor (*Adora1* or *Adora3*), but predominantly expressed the A2b and A2a receptors (*Adora2b* and *Adora2a*), respectively (*Figure 2E*). Accordingly, an A2b antagonist (alloxazine) inhibited the apoptotic cell supernatant-induced *Thbs1* gene expression in BMDMs, but only weakly inhibited the induced Thbs1 gene expression in thio-pMacs (*Figure 2F*). An A2a antagonist (SCH58261) had an inhibitory effect on thio-pMacs, but not on BMDMs, whereas antagonists of A1 (DPCPX) and A3 (VUF5574) had no effect on thio-pMacs or BMDMs. *Adora2a*$^{-/-}$ thio-pMacs lost their ability to upregulate the expression of *Nr4a1* and *Thbs1* genes in response to the apoptotic cell supernatant (*Figure 2G*), confirming the essential role of the A2a receptor in this process. These results confirmed that adenine nucleotides released from apoptotic cells were converted to adenosine, which in turn activated macrophages by binding to adenosine receptors.

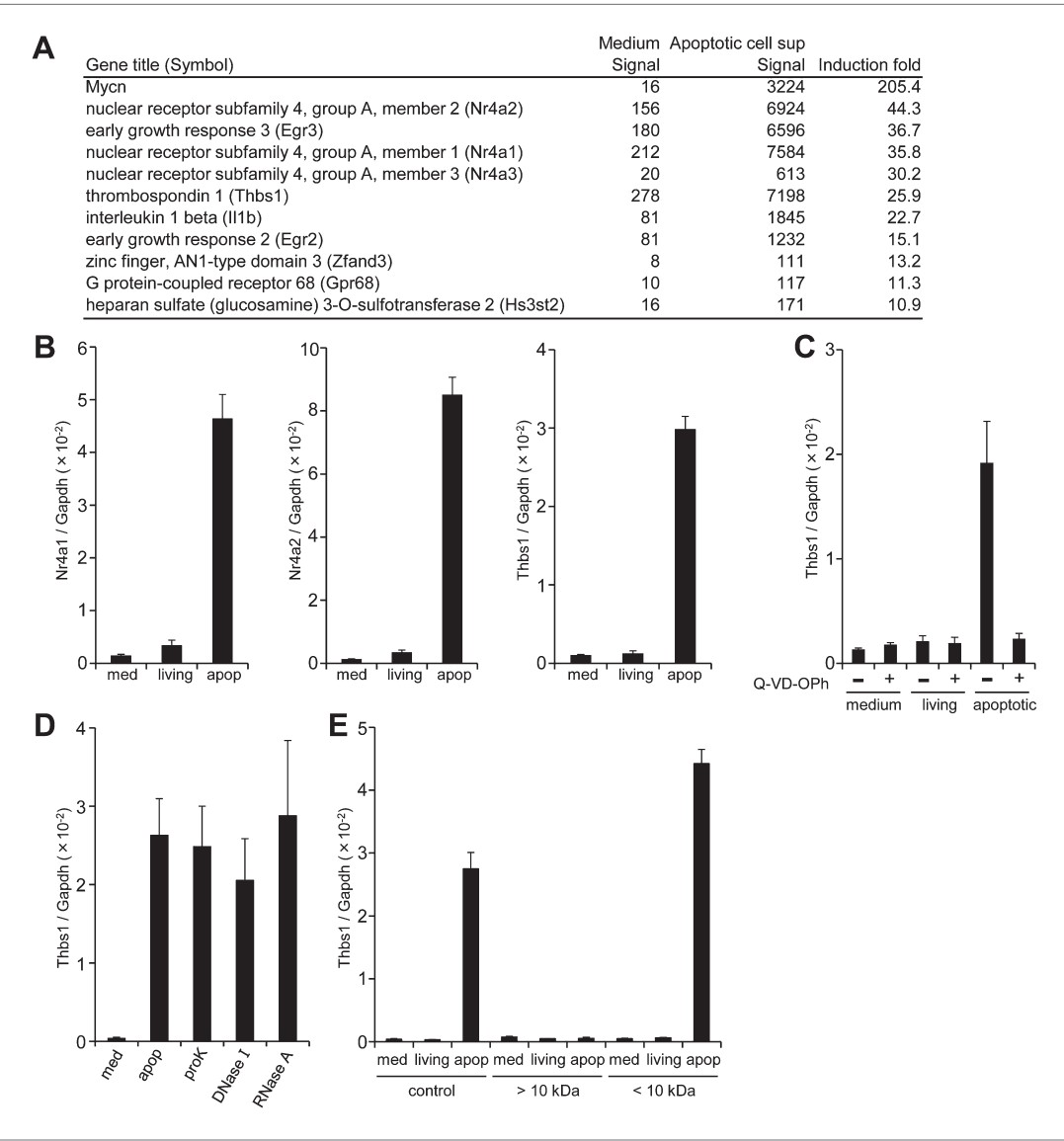

**Figure 1**. Factor(s) released from apoptotic cells stimulate gene expression in macrophages. (**A** and **B**) BMDMs were incubated for 1 hr with medium or with the supernatant of W3 cells that had been treated with (apoptotic) or without (living) 30 units/ml FasL. RNA from BMDMs was then subjected to microarray analysis. (**A**) Genes whose expression was upregulated more than 10-fold after incubation with the apoptotic cell supernatant are listed. (**B**) *Nr4a1*, *Nr4a2*, and *Thbs1* mRNA levels were quantified by real-time RT-PCR, and normalized to *Gapdh* mRNA. (**C**) W3 cells were pre-treated with or without 20 μM Q-VD-OPh for 20 min and stimulated with or without 30 units/ml FasL. BMDMs were then incubated for 1 hr with the supernatant of Q-VD-OPh-treated (+) or untreated (−) living or FasL-treated apoptotic W3 cells, and *Thbs1* mRNA levels were determined by real-time RT-PCR. (**D**) BMDMs were incubated with the supernatant of apoptotic W3 cells that had been treated with proteinase K (proK), DNase I or RNase A, and *Thbs1* mRNA levels were determined. (**E**) Medium, the culture supernatant of healthy W3 cells (living) or apoptotic W3 cells (apop) were subjected to ultrafiltration through a 10 kDa-cutoff filter, and the filtrate (<10 kDa) and concentrate (>10 kDa) were tested for their ability to induce *Thbs1* expression in BMDMs. Experiments were performed in triplicates, and the average values are plotted with SD (bars). All experiments were repeated at least twice with BMDM from different mice, and representative data are shown.

The following figure supplements are available for figure 1:

**Figure supplement 1**. FasL-induced apoptosis in W3 cells.

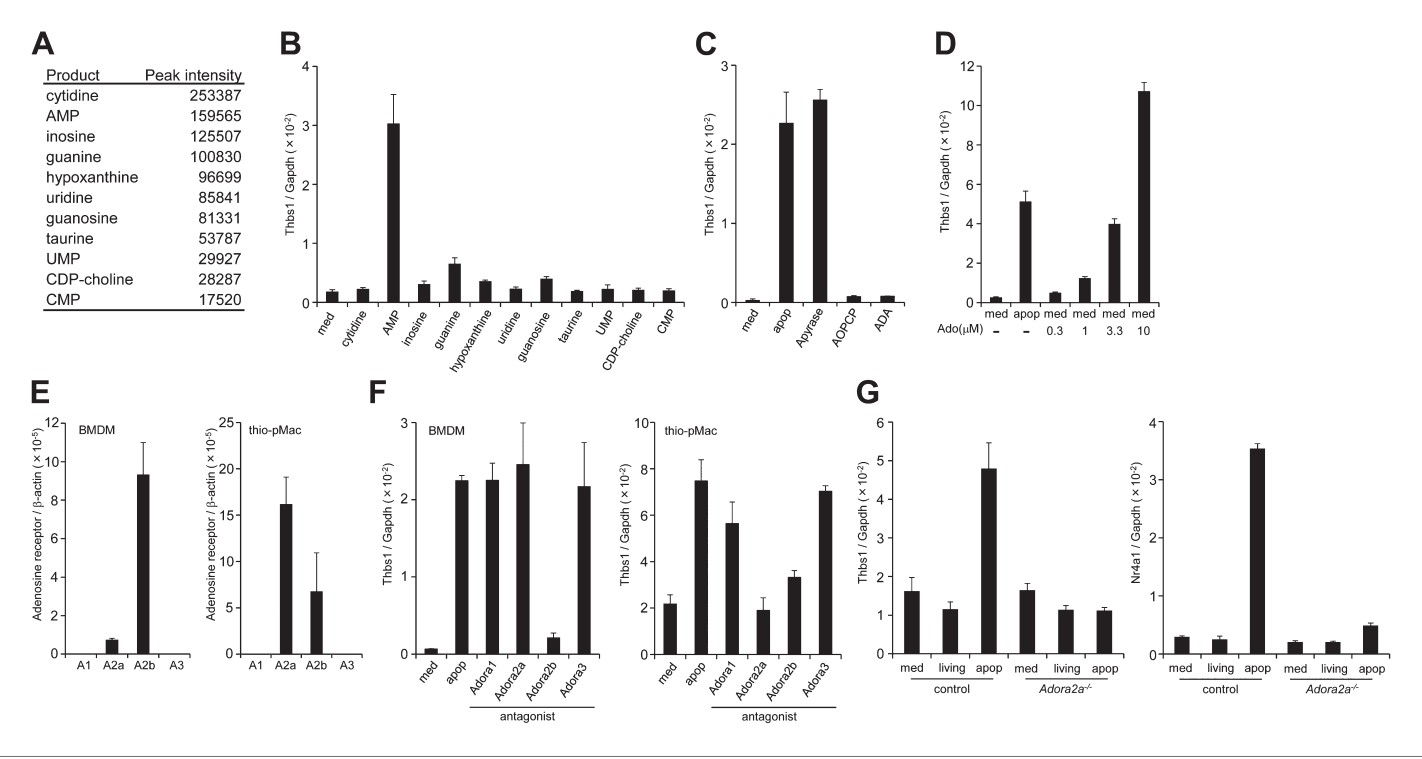

**Figure 2**. Identification of the factor in apoptotic cell supernatant that stimulates macrophage gene expression. (**A**) The supernatant of apoptotic W3 cells was subjected to LC-MS analysis. The relative concentration of each compound is represented by the base peak intensity. (**B**) BMDMs were incubated with medium containing 10 μM of the indicated reagents for 1 hr, and *Thbs1* mRNA levels were determined by real-time RT-PCR. (**C**) Apoptotic W3 cell supernatant was pretreated with 25 mU/ml apyrase at 37°C for 1 hr. BMDMs were incubated for 1 hr with the pretreated or untreated supernatant, and the *Thbs1* mRNA levels were determined. BMDMs were also treated with the supernatant in the presence of 10 μM AOPCP or 0.9 U/ml adenosine deaminase (ADA), and the *Thbs1* mRNA levels were quantified as above. (**D**) BMDMs were incubated with apoptotic cell supernatant (apop) or medium supplemented with the indicated concentrations of adenosine (Ado), and the *Thbs1* mRNA levels were determined. (**E**) The mRNA levels of the *Adora1*, *Adora2a*, *Adora2b*, and *Adora3* expressed in BMDMs and thio-pMacs were determined by real-time RT-PCR, and normalized to *β-actin* mRNA. (**F**) BMDMs and thio-pMacs were incubated with W3 apoptotic cell supernatant and adenosine receptor antagonists, 5 nM 8-cyclopentyl-1,3-dipropylxanthine (DPCPX) (A1), 10 nM SCH58261 (A2a), 5 μM alloxazine (A2b), or 130 nM VUF5574 (A3), and *Thbs1* mRNA levels were determined. (**G**), Thio-pMacs from *Adora2a^{+/+}* (control) or *Adora2a^{-/-}* mice were incubated with medium, apoptotic or living W3 cell supernatants, and the *Nr4a1* and *Thbs1* mRNA levels were determined by real-time RT-PCR. Experiments were performed in triplicate, and the average values are plotted with SD (bars). All experiments were repeated at least twice with BMDM or thio-pMacs from different mice, and representative data are shown.

## Caspase-dependent accumulation of AMP

Apoptotic cells are reported to release ATP (***Elliott et al., 2009***), and AMP can be generated from ATP by ectonucleotide pyrophosphatase/phosphodiesterase (E-NTPDase) (***Bours et al., 2006***). To examine whether these events were involved in the apoptotic supernatant activity, the release of adenine nucleotides from apoptotic cells was followed over time. W3 cells were treated with FasL, and aliquots of the culture supernatant were collected at different time points and analyzed by LC-MS. As shown in ***Figure 3A***, healthy cells secreted low levels of AMP, whereas FasL-induced apoptosis caused a gradual increase of AMP in the supernatant. At 2 hr after the addition of FasL, the AMP level had increased to more than 4 μM in a culture of $1.0 \times 10^7$ cells/ml. The concentration of ADP also increased, but was less pronounced compared with that of AMP. In contrast, at 2 hr the concentration of ATP was substantially lower, 40 nM or 40 pmoles per $10^7$ cells. In various apoptosis systems, the intracellular ATP level is reported to remain high (***Eguchi et al., 1997***; ***Bossy-Wetzel et al., 1998***). However, the intracellular concentration of ATP in the FasL-treated W3 cells rapidly decreased, whereas the concentrations of ADP and AMP concomitantly increased (***Figure 3B***), suggesting that AMP was directly released from apoptotic cells. Exogenously added ATP was slowly degraded in the supernatant of FasL-treated W3 cells (***Figure 3—figure supplement 1***) providing additional support that AMP was released directly

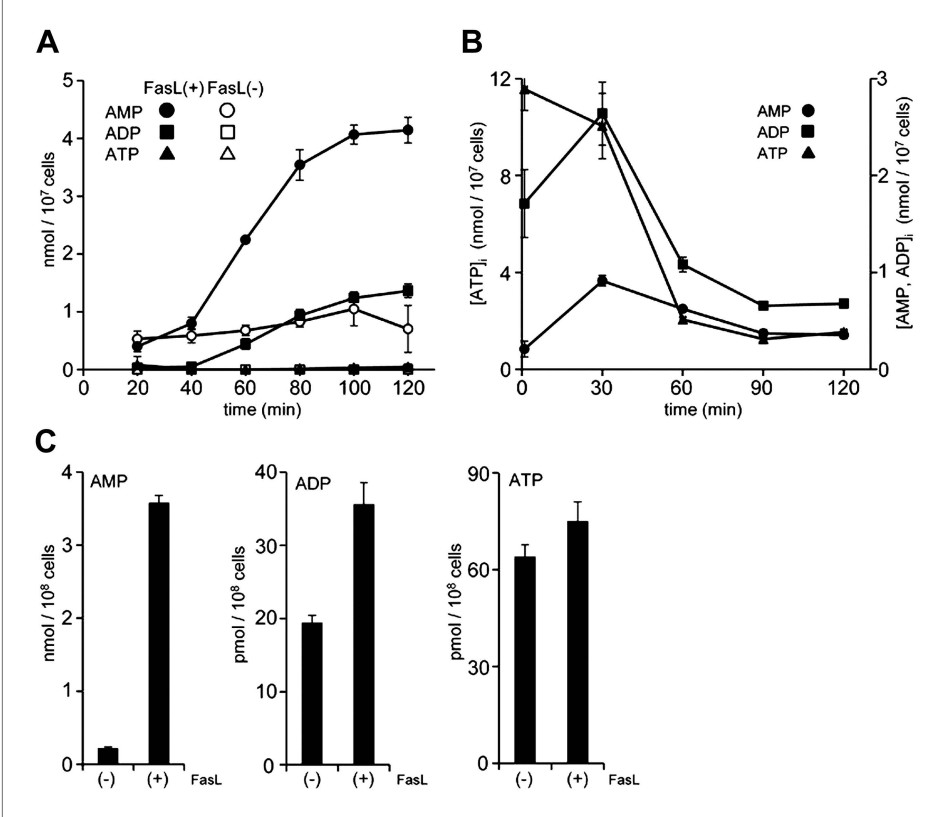

**Figure 3**. Release of adenine nucleotides from apoptotic cells. (**A** and **B**) W3 cells were treated with (●,■,▲) or without (○,□,△) 120 units/ml FasL, and the concentrations of AMP (○,●), ADP (□,■) and ATP (△,▲) in the supernatant (**A**) and cells (**B**) were determined by LC-MS or HPLC analysis at the indicated times. (**C**) Mouse thymocytes were treated with (+) or without (−) 120 units/ml FasL for 90 min, and the concentrations of AMP, ADP, and ATP in the supernatants were determined. Experiments were performed in triplicate, and the average values are plotted with SD (bars). The treatment of cells with FasL and quantification of adenine nucleotides were repeated at least twice, and representative data are shown.

The following figure supplements are available for figure 3:

**Figure supplement 1**. Minimal degradation of ATP in the apoptotic cell supernatant.

from the apoptotic cells, and not extracellularly converted from ATP. When mouse thymocytes in a culture of $1 \times 10^8$ cells/ml were treated with FasL for 90 min to induce apoptosis in approximately 70% of the cells (8% were PI positive), the culture supernatant was found to contain 3.5 μM AMP, whereas the supernatant of healthy cells cultured for 1 hr contained approximately 0.2 μM AMP (**Figure 3C**). Consistent with our finding using W3 cells, the levels of ATP and ADP in the FasL-treated thymocyte were 40 and 100 times lower than that of AMP, respectively. These results further suggest that AMP was directly released from primary thymocytes undergoing apoptosis.

## Pannexin 1-dependent release of AMP from apoptotic cells

The AMP release from FasL-treated W3 cells was suppressed by a caspase inhibitor, Q-VD-OPh, in a dose-dependent manner (**Figure 4A**). **Chekeni et al. (2010)** reported that Pannexin 1 is cleaved by caspase in apoptotic cells and mediates the release of ATP from these cells. Pannexins are a family of plasma membrane channel proteins consisting of 3 members (**Baranova et al., 2004**). W3 cells and thymocytes expressed *Panx1*, but not *Panx2* or *Panx3* (**Figure 4—figure supplement 1**). A synthetic derivative of glycyrrhetinic acid (CBX, carbenoxolone), an inhibitor of Pannexin and Connexin channels (**D'Hondt et al., 2009**), inhibited the release of adenine nucleotides (AMP, ADP, and ATP) from FasL-treated W3 cells, whereas gadolinium chloride (GdCl₃), an inhibitor of the maxi-anion channel, failed

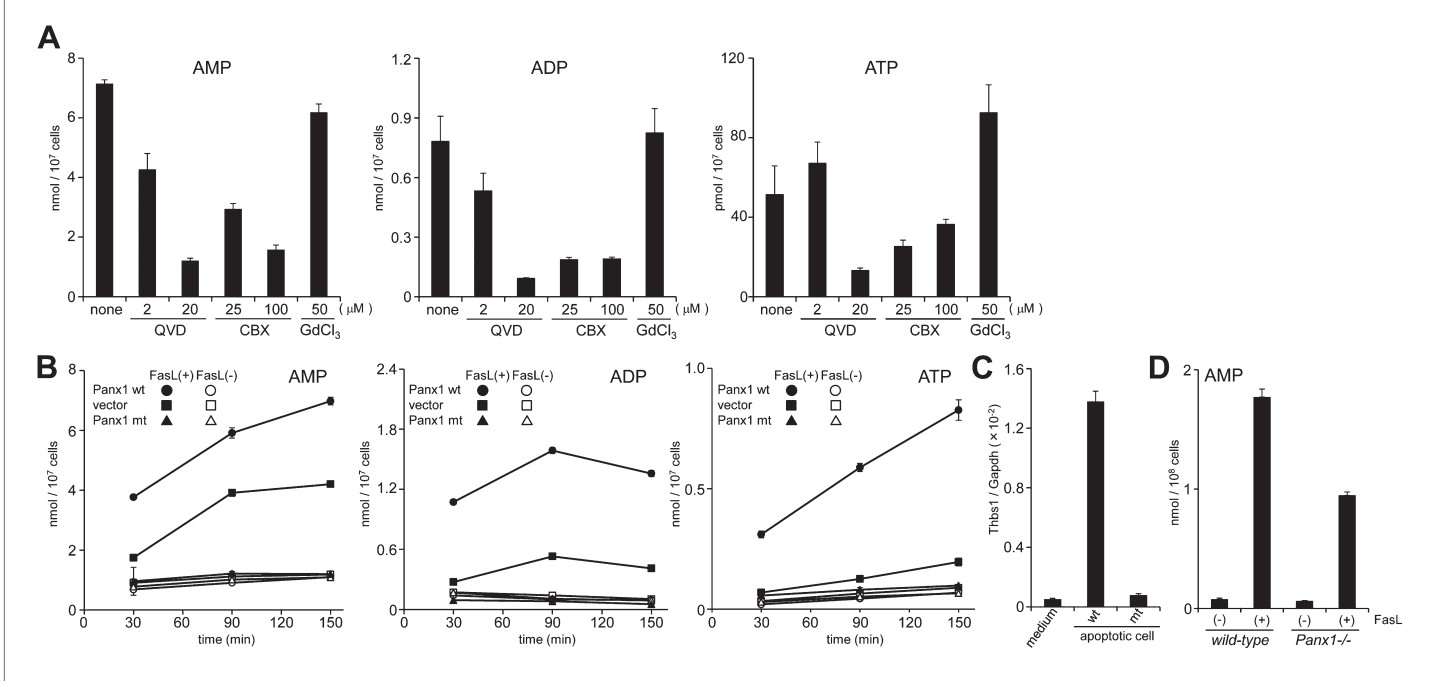

**Figure 4**. Pannexin 1-dependent release of adenine nucleotides from apoptotic cells. (**A**) After pre-treatment with the indicated concentrations of Q-VD-OPh (QVD), carbenoxolone (CBX), or $GdCl_3$ for 20 min, W3 cells were incubated with 120 units/ml FasL for 90 min. The AMP, ADP, and ATP concentrations in the supernatants were determined by LC-MS. (**B**) W3 cells transformed with empty pMXs vector (■,□), or with vectors bearing wild-type (wt) (●,○) or caspase-resistant mutant (mt) (▲,△) Pannexin 1 were treated (■,●,▲) or not treated (□,○,△) with 120 units/ml FasL, and the concentrations of adenine nucleotides in the supernatants were determined at the indicated times. (**C**) The parental W3 cells (wt) and W3 cell transformants expressing caspase-resistant form of pannexin 1 (mt) were treated with 30 units/ml FasL for 30 min, added to BMDMs, and incubated for 1 hr. The *Thbs1* mRNA level in BMDMs was then determined by real-time RT-PCR. (**D**) Thymocytes from *Panx1+/+* or *Panx1−/−* mice were incubated with (+) or without (−) 120 units/ml FasL, and the concentration of adenine nucleotides in the supernatants were determined at the indicated times. Experiments were performed in triplicate, and the average values are plotted with SD (bars). All experiments were repeated at least twice, and the representative data are shown.

The following figure supplements are available for figure 4:

**Figure supplement 1**. *Panx* mRNA expression in W3 cells and mouse thymocytes.

**Figure supplement 2**. Expression of wild-type and caspase-resistant Pannexin 1 in W3 cell transformants.

to inhibit their release from the apoptotic cells (*Figure 4A*). Mouse Pannexin 1 contains a caspase 3 recognition sequence (Asp-Ile-Ile-Asp) at amino acids 375 to 378. Therefore, we established W3 cells expressing wild-type or caspase-resistant Pannexin1 (*Figure 4—figure supplement 2*). As shown in *Figure 4B*, W3 transformants expressing wild-type Pannexin 1 released significantly higher levels of adenine nucleotides than the parental W3 cells upon FasL treatment. The caspase 3 recognition sequence of Pannexin 1 was then mutated to a caspase-resistant sequence (Ala-Ile-Ile-Ala). When W3 cells were transformed with the caspase-resistant form of Pannexin 1, the transformants completely lost their ability to release adenine nucleotides following FasL treatment (*Figure 4B*), indicating that the caspase cleavage of Pannexin 1 was required for the release of adenine nucleotides. Co-culturing BMDM with FasL-treated W3 cells upregulated the *Thbs1* gene expression in BMDMs (*Figure 4C*), this was not observed when the W3 cells were transformed with the caspase-resistant form of pannexin1. Furthermore, *Panx1−/−* thymocytes exhibited reduced AMP release following FasL treatment compared with similarly treated wild-type thymocytes (*Figure 4D*). Incomplete inhibition of the AMP release by the *Panx1−/−* thymocytes may have been due to the expression of connexins in mouse thymocytes, or because a significant fraction of the thymocytes had undergone secondary necrosis (data not shown). These results indicated that caspase activation resulted in the opening of Pannexin 1 channels, leading to AMP release.

## Adenosine-mediated immunosuppression

The above results suggested that the AMP released from apoptotic cells is converted to adenosine by macrophages, and stimulates a set of genes in macrophages via activation of adenosine receptor. Among adenosine receptors, the A2 receptors are known to exert an anti-inflammatory effect on macrophages (*Sitkovsky and Ohta, 2005*). To confirm that AMP could exert a similar effect on macrophages, mouse resident peritoneal macrophages were treated with zymosan, a glucan that stimulates macrophages via TLR2 (*Takeuchi et al., 1999*), in the presence or absence of AMP. As shown in *Figure 5A*, zymosan-stimulated macrophages to produce TNFα, a classic pro-inflammatory cytokine, and MIP-2 (CXCL-2), a critical chemokine for neutrophil migration. The production of these factors was significantly suppressed by the presence of AMP.

We next examined the role of AMP secretion from apoptotic cells using the zymosan-induced peritonitis model (*Cash et al., 2009*). 3 hr after the injection of zymosan particles into the mouse peritoneal cavity, 30% of the peritoneal cells were Ly6B.2 and Ly6G double positive, and 20% of the double-positive cells were Annexin V positive, indicating that neutrophils had infiltrated into the cavities and undergone apoptosis (*Figure 5B*). 2 hr after the injection of zymosan-particles, the levels of TNFα and MIP-2 in the peritoneal lavage were comparable between the wild-type and $Panx1^{-/-}$ mice. However, by 6 hr the levels of these inflammatory mediators in the peritoneal cavity remained high in $Panx1^{-/-}$ mice, but had returned to normal in the wild-type mice (*Figure 5C*). These results indicated that Pannexin 1 was required to quickly resolve the zymosan-induced peritonitis. Consistent with the results from $Panx1^{-/-}$ mice, the levels of inflammatory mediators in $Adora2a^{-/-}$ peritoneal lavage fluids were greater than those in the $Adora2a^{+/+}$ or $Adora2a^{+/-}$ lavage at 6 hr after zymosan injection (*Figure 5D*). Taken together, these results suggested that apoptotic neutrophils released adenosine nucleotides, most likely AMP, through Pannexin 1 channels during acute peritoneal inflammation. The secreted AMP was, in turn, converted to adenosine by macrophages, leading to the activation of adenosine A2a receptor and the upregulation of *Nr4a* and *Thbs1* gene expression, ensuring the rapid resolution of peritonitis.

## Discussion

Adenine nucleotides act not only as intracellular energy carriers but also as extracellular signaling molecules (*Bours et al., 2006*). For instance, extracellular ATP regulates the microglial response to brain injury via P1 or P2 purinergic receptors (*Davalos et al., 2005*), and adenosine modulates sleep–wake cycles via A2a adenosine receptor (*Urade et al., 2003*). Although the release of adenine nucleotides mainly occurs by exocytosis (*Pascual et al., 2005*), it is also mediated by transport via channel proteins. Connexins, which form gap junctions, can serve as hemichannels, and allow ATP to exit (*Cotrina et al., 1998*; *Spray et al., 2006*). Pannexins, which carry 4 transmembrane regions with intracellular N- and C-terminal tails, are structurally similar to connexins and also form hemichannels (*Yen and Saier, 2007*). Various physiological and pathological stimuli such as $Ca^{2+}$-influx, shear stress, and apoptotic cell death, can cause Pannexin 1 channels to open, allowing the passage of molecules smaller than 1.0 kDa (*D'Hondt et al., 2009*). *Chekeni et al. (2010)* recently showed that in the early stages of apoptosis, caspases cleave Pannexin 1 resulting in the release of ATP, which can serve as a 'find me' signal to attract macrophages.

In this study, we found that AMP was present at 30- to 100-fold higher concentrations than ATP in the culture supernatants of apoptotic thymocytes and a T cell line. The ATP concentration in healthy cells is approximately 10–50 times higher than that of AMP. However, when cells undergo apoptosis, ATP is quickly hydrolyzed to AMP, while its generating system is inactivated by caspases (*Ricci et al., 2004*). The caspase-cleaved Pannexin channel also contributes to the cellular loss of ATP by allowing ATP to exit cells through the plasma membrane. Thus, the AMP found in apoptotic cell supernatants could have been extracellularly converted from ATP by E-NTPDase. However, the ability of apoptotic or healthy W3 cell supernatants to hydrolyze ATP was very low, suggesting that E-NTPDase was not responsible for the accumulation of AMP in the apoptotic cell supernatants. Thus, it is possible that AMP generated intracellularly in apoptotic cells was released from through these channels.

The engulfment of apoptotic cells is a silent process that does not activate inflammation (*Nagata et al., 2010*; *Ravichandran, 2011*). Since macrophages ingesting apoptotic cells produce TGFβ or IL-10 that inhibits the production of inflammatory cytokines (*Voll et al., 1997*; *Fadok et al., 1998*; *Lucas et al., 2006*), this interaction and/or engulfment was thought to be responsible for the

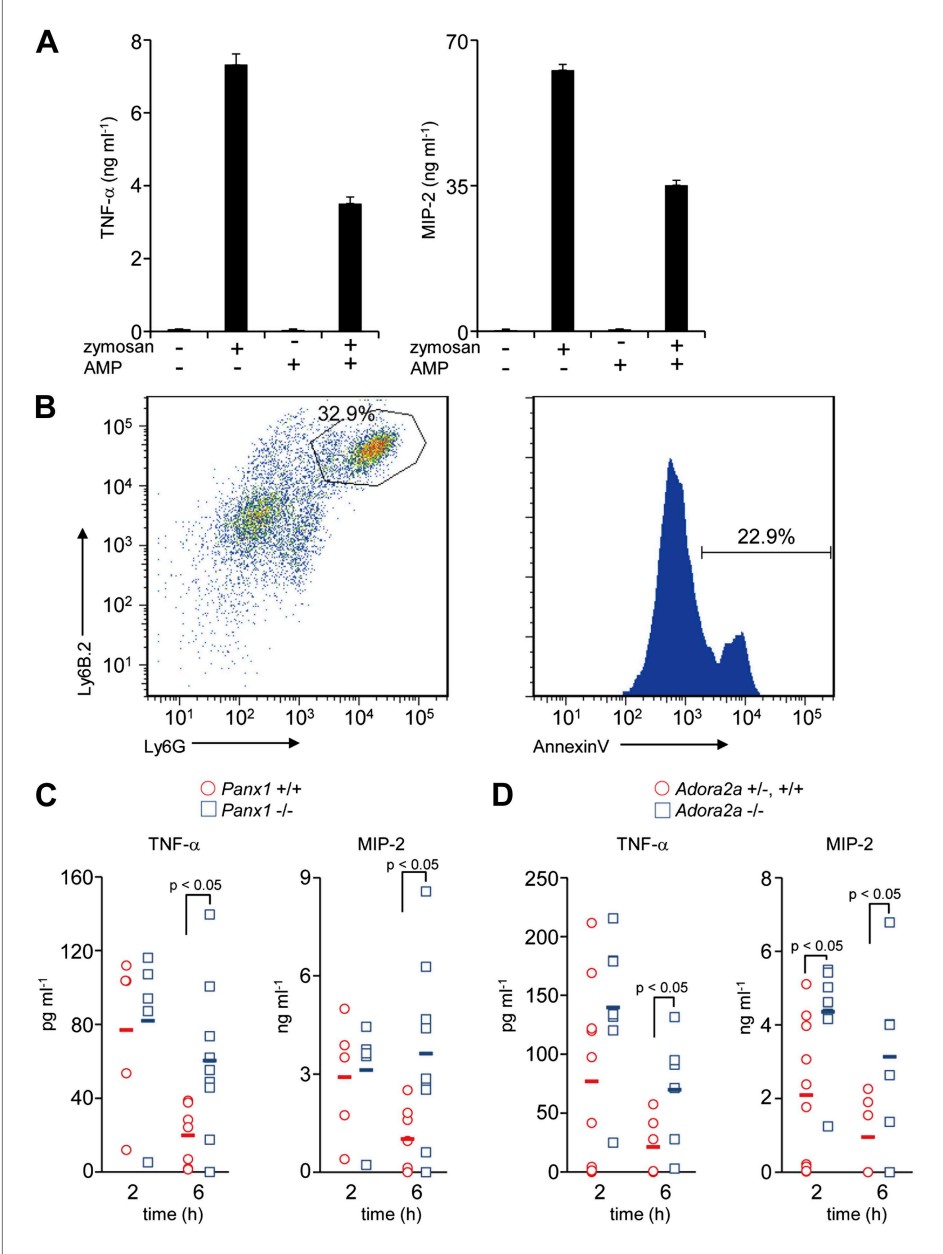

**Figure 5**. Immunosuppressive function of Pannexin 1 and A2a adenosine receptor in a mouse peritonitis model. (**A**) Mouse resident peritoneal macrophages were incubated with 50 µg/ml zymosan A in the presence (+) or absence (−) of 100 µM AMP for 16 hr. The TNFα and MIP-2 levels in the culture supernatant were determined in triplicate by ELISA, and the average values are plotted with S.D. (bars). The experiments were performed at least twice, and representative data are shown. (**B**) Peritoneal cells were collected at 3 hr after the injection of zymosan A (250 mg/kg), and stained with anti-Ly6B.2 and anti-Ly6G antibodies (left panel). Some of the Ly6B.2 and Ly6G double-positive cells (circled in the left panel) were stained with Cy5-labeled Annexin V (right panel). (**C** and **D**) After zymosan injection, peritoneal lavage fluid was collected at 2 hr (n = 5) or 6 hr (n = 7) of $Panx1^{+/+}$, and at 2 hr (n = 5) or 6 hr (n = 10) of $Panx1^{-/-}$ littermates (**C**), or 2 hr (n = 10) or 6 hr (n = 6) of $Adora2a^{+/+\ or\ +/-}$, and at 2 hr (n = 8) or 6 hr (n = 6) of $Adora2a^{-/-}$ mice (**D**). The TNFα and MIP-2 levels in the fluids were determined by ELISA. The Student's $t$ test was used for the statistical analysis, and the p values are shown.

anti-inflammatory effect. Furthermore, *Köröskényi et al. (2011)* recently reported that macrophages engulfing apoptotic cells released adenosine to suppress inflammation. In contrast, we show here that apoptotic cells secrete AMP, which is converted to adenosine, resulting in macrophage activation via

adenosine receptors, followed by the upregulation of *Thbs1* and *Nr4a* gene expression. When the apoptotic cells could not release AMP due to the expression of caspase-resistant form of pannexin 1, the *Thbs1* gene expression was not up-regulated. Furthermore, the AMP released from the resident peritoneal macrophages engulfing apoptotic cells was significantly smaller than that released from apoptotic cells (HY, and SN, data not shown). These results suggest that the contribution of the inter-action of apoptotic cells with macrophages to anti-inflammatory effect may not be great. Thbs1 is the major activator of TGFβ (*Crawford et al., 1998*), whereas Nr4a family members inhibit the expression of proinflammatory cytokines such as TNFα, IL-8, and IL-6 in macrophages (*Bonta et al., 2006*), by recruiting a repressor complex to their promoter (*Saijo et al., 2009*). Thus, this may be one mechanism that suppresses inflammation during the apoptosis process.

The conversion of AMP to adenosine is catalyzed by CD73, an ecto-5'-nucleotidase, expressed on the surface of endothelial and immune cells (*Deaglio and Robson, 2011*). CD73-null mice spontane-ously develop autoimmune diseases such as glomerulitis and peritubular capillaritis, accompanied by enhanced inflammation (*Blume et al., 2012*). They are also susceptible to polymicrobial sepsis induced by cecal ligation and puncture, and exhibit increased mortality and systemic inflammation (*Hasko et al., 2011*). Our results suggest that the substrate for CD73 in these models may be AMP released from apoptotic cells. If AMP cannot be converted to adenosine in the absence CD73, then activa-tion of the immune system in response to the 'danger signal(s)' released from apoptotic cells would not be opposed.

We conclude that apoptotic cells themselves contribute to the 'anti-inflammatory' nature of the engulfment process by secreting AMP as an immunosuppressive 'calm-down' signal. Depending on the types of cells and apoptotic stimuli, intracellular ATP levels remain high (*Bossy-Wetzel et al., 1998*; *Eguchi et al., 1997*; *Zamaraeva et al., 2005*) or rapidly decreases (*Gleiss et al., 2002*; *Vander Heiden et al., 1999*) during apoptosis. It will be interesting to study whether this mechanism impacts various types of inflammation that are accompanied by apoptosis.

## Material and methods

### Mice and reagents

C57BL/6J mice were purchased from Japan SLC. *Adenosine receptor A2a (Adora2a)* knockout mice (*Chen et al., 1999*) were from Dr Jian-Fan Chen (Boston University School of Medicine). *Pannexin 1* knockout mice (*Panx1tm1a(KOMP)Wtsi*) were purchased from the University of California at Davis' KOMP Repository (http://www.komp.org/). All mice were housed in a specific pathogen-free facility at Kyoto University, Graduate School of Medicine. The committee of Kyoto University, Graduate School of Medicine, approved our study (Reference Number: Med Kyo 12,107). The leucine-zipper tagged soluble human Fas ligand (FasL) was prepared as described previously (*Shiraishi et al., 2004*). One unit of FasL was defined as the dilution that gives a half-maximum response with W3 cells, mouse WR19L cells expressing mouse Fas (*Watanabe-Fukunaga et al., 1992*).

Adenosine deaminase, apyrase, DNase I, RNase A and proteinase K were purchased from Calbiochem (Germany), New England BioLabs (Ipswich, MA), Takara Bio (Japan), Ambion (Austin, TK), and Merck (Germany), respectively. A caspase inhibitor, N-(2-Quinolyl)valyl-aspartyl-(2,6-difluorophenoxy) methyl Ketone (Q-VD-OPh) was from R&D Systems (Minneapolis, MN). Adenosine, AMP, ADP, and ATP were from Nacalai (Japan). Cytidine, inosine, guanine, hypoxanthine, uridine, guanosine, taurine, UMP, CDP-choline CMP, and α, β-methyleneadenosine 5'-diphosphate (AOPCP) were from Sigma (St. Louis, MO). Antagonists for A1 (8-cyclopentyi-1,3-dipropylxanthine), A2a (SCH58261), A2b (Alloxazine), and A3 Adenosine receptor (VUF5574) were from Sigma. Carbenoxolone, an inhibitor of Pannexin channel, was purchased from Sigma. Gadolinium chloride hexahydrate ($GdCl_3 \cdot 6H_2O$), an inhibitor of maxi anion channel, was from Aldrich (St. Louis, MO).

### Preparation of macrophages

For mouse bone marrow-derived macrophages (BMDMs), bone marrow cells from female C57BL/6J mice at 8 weeks of age were treated with Buffer EL (Qiagen) to remove red blood cells, and cultured for more than 7 days with DMEM containing 10% FCS supplemented with 5% supernatant of CMG14-12 cells producing mouse M-CSF (*Takeshita et al., 2000*). For thioglycollate-elicited peritoneal macro-phages (thio-pMacs), mice were injected i.p. with 3% thioglycollate (Sigma). Peritoneal excudate cells were obtained 4 days later, and cultured in DMEM containing 10% FCS. For resident peritoneal

macrophages (rpMacs), mouse peritoneal cells were cultured at 37°C for 2 hr in RPMI1640 containing 10% FCS on 12-well plates. Adherent cells were washed once with fresh medium, and used as rpMac.

## Treatment of macrophages with apoptotic cells or cell supernatant

W3 cells were treated at $2 \times 10^6$ cells/ml with human FasL at 37°C for 30 min, washed, re-suspended at a concentration of $1 \times 10^7$ cells/ml with RPMI containing 1% FCS, and further incubated for 60 min at 37°C. At this stage, 94% cells were Annexin V-positive and PI-negative. The same treatment without FasL generated about 3% Annexin V-positive cells. The culture supernatant of W3 cells was recovered, 0.5 ml-aliquots were added to mouse BMDMs ($2 \times 10^5$ cells) or thio-pMacs ($5 \times 10^5$ cells) in a 12-well microtiter plate. To study the effect of apoptotic cells, the FasL-treated W3 cells ($5 \times 10^6$) were washed, re-suspended in RPMI containing 1% FCS at a concentration of $1 \times 10^7$ cells/ml, and added to BMDMs. After incubation at 37°C for 1 hr, the macrophages were washed with PBS, and total RNA for RT-PCR was prepared by using RNeasy mini kit (Qiagen). To study the effect of zymosan on the gene expression, rpMacs ($6 \times 10^5$ cells) in a 12-well plate were treated with 50 μg/ml of zymosan and 100 μM AMP for 16 hr.

The fractionation of the supernatant of apoptotic W3 cells was carried out with a 10 kDa cutoff (MWCO) ultrafilter (Amicon Ultra, Millipore).

## Microarray analysis, real time RT-PCR, and ELISA

For microarray analysis, RNA was amplified and labeled using GeneChip 3' IVT Express Kit (Affymetrix, Santa Clare, CA), and the labeled probe was used for hybridization to a Mouse Genome 430 2.0 Array GeneChip (Affymetrix). The hybridized signals were detected with an Affymetrix Agilent Microarray scanner, and the array image was analyzed with Microarray Analysis Suit Version 5.0 (Affymetrix).

Real-time RT-PCR was performed with LightCycler 480 instrument (Roche, Indianapolis, IN) after total RNA was reverse transcribed by High Capacity RNA-to-cDNA kit (Applied Biosystems, Foster City, CA). Following primers were used: 5'-ATGCCTCCCCTACCAATCTTC-3' and 5'-CACCAGTTCCT-GGAACTTGGA-3' for *Nr4a1*; 5'-TCACCTCCGGTGAGTCTGATC-3' and 5'-TGCTGGATATGTTGGG-TATCATCT-3' for *Nr4a2*; 5'-GGTGCTGCAGAATGTGAGGT-3' and 5'-GCTGGAACCGTTCACCACGT-3' for *Thbs1*; 5'-AGCAGGCATCTGAGGGCCCA-3' and 5'-GAGAGCAATGCCAGCCCCGG -3' for *Gapdh*; 5'-TGTTACCAACTGGGACGACA-3' and 5'-CCATCACAATGCCTGTGGTA -3' for *β-actin*; 5'-CATTGGGCCACAGACCTACT-3' and 5'-CAAGGGAGAGAATCCAGCAG-3' for *Adora1*; 5'-CTCTT-CTTCGCCTGCTTTGT-3' and 5'-AATGACAGCACCCAGCAAAT-3' for *Adora2a*; 5'-CCTTTGCCATC-ACCATCAG-3' and 5'-GTGTCCCAGTGACCAAACCT-3' for *Adora2b*; 5'-TCAGCCTGCAAGTCAAGATG-3' and 5'-CAGCAAAGGCCCAAGAATAG-3' for *Adora3*; 5'-GCCAAGAGTGCTCGAGATTT-3' and 5'-TTC-AGGACGCCTGATTTGAT-3' for *Panx1*; 5'-CAAGAAGGCCAAGACTGAGG-3' and 5'-CAGGATGGTGAG-AGGGTCAG-3' for *Panx2*; 5'-CCTCAGCTCTGACCTGCTGT-3' and 5'-GAGGAGTAAGAGGGCGTTCC-3' for *Panx3*. The amount of specific mRNA was normalized to *Gapdh* or *β-actin* mRNA for each individual sample.

TNFα and MIP-2 levels were measured by ELISA kit from BioLegend and R&D Systems, respectively, according to the manufacture's instruction.

## Identification of metabolites in the apoptotic cell supernatant

W3 cells ($2 \times 10^6$ cells/ml) were treated with 30 units/ml human FasL for 30 min, washed, re-suspended at a concentration of $1 \times 10^7$ cells/ml in serum-free RPMI medium supplemented with ITS-X (insulin, transferrin and selenous acid; Invitrogen, Carlsbad, CA), and further incubated for 1 hr. The supernatant was subjected to LC-MS analysis in Kazusa DNA Research Institute (Japan), using an Agilent 1200 liquid chromatography system (Agilent technologies, Santa Clare, CA) connected to a Finnigan LTQ Orbitrap XL mass spectrometer (Thermo Fisher Scientific, Waltham, MA). The chromatography was performed at a constant flow rate of 0.5 ml/min with 3–95% linear gradient of acetonitrile in 0.1% formic acid, using a TOSOH TSKgel ODS-100V column (TOSOH, Japan).

Data were analyzed according to a molecular formula calculation in following databases: ExactMassDB (http://webs2.kazusa.or.jp/mfsearcher/exmassdb/), KEGG (http://www.genome.jp/kegg/), KNApSAcK (http://kanaya.naist.jp/KNApSAcK/), Flavonoid Viewer (http://www.metabolome.jp/software/FlavonoidViewer/) and LipidMAPS (http://www.lipidmaps.org/).

## Quantification of adenosine and adenine nucleotides by LC-MS

For quantification of extracellular or intracellular adenine nucleotides, W3 cells ($1 \times 10^7$ cells/ml) or thymocytes ($1 \times 10^8$ cells/ml) from 6- to 8-week-old mice were treated with 120 units/ml human

FasL at 37°C in RPMI containing 1% FCS. After centrifugation at 1700×*g* for 4 min, the supernatant of the thymocytes was extracted by chloroform and methanol to remove lipids and proteins, lyophilized and reconstituted with distilled water. The LC-MS analysis was performed using a LC-MS system (Alliance/ZQ-4000, Waters, Milford, MA). Samples were separated at 45°C with Cosmosil-PAQ column (2.0 mm × 250 mm; Nacalai) using 1–45% acetonitrile gradient at a flow rate of 0.2 ml/min, and compounds were detected using negative electrospray ionization. The concentration of adenosine, AMP, ADP and ATP was determined by comparing the peak areas with those of standards. In some cases, adenine nucleotides were quantified by HPLC using a Gilson PLC2020 system equipped with a COSMOSIL-PAQ column (Nacalai). The HPLC was performed using acetonitrile gradient (1–45% acetonitrile in 30 min) in the presence of 1 mM dihexyl ammonium acetate (Tokyo Chemical Industry, Japan) at a flow rate of 1 ml/min. ATP level was also quantified by luciferase assay system.

## Transformation of mouse W3 cells with Pannexin 1-expression vector

A full-length cDNA for mouse *Panx1* (GenBank accession number NM_019482.2) was prepared by RT-PCR from mouse BMDMs, and its authenticity was verified by DNA sequencing. The caspase-recognition sequence, Asp-Ile-Ile-Asp, at amino acid position 375–378 in mouse Panx1 was mutated by recombinant PCR to Ala-Ile-Ile-Ala. The cDNA was Flag-tagged at the C-terminus, and introduced into the pMXs-puro retroviral vector (*Kitamura et al., 2003*). Human 293T cells were co-transfected with the pMXs-puro expression vector, together with pEF-gag-pol and pE-Ampho (Takara Bio). 2 days later, the amphotropic retrovirus in the culture medium was concentrated by centrifugation at 6000×*g* for 16 hr, and used to infect W3 cells in the presence of 10 µg/ml polybrene. Transformants were selected in medium containing 1.0 µg/ml puromycin. To confirm the expression of mouse Panx1, the cells were permeabilized with 0.3% saponin, and stained with FITC-conjugated anti-Flag M2 (Sigma), followed by a flow cytometry with FACSCalibur (BD Bioscience, San Jose, CA).

## Zymosan-induced peritonitis

Zymosan-induced peritonitis was performed essentially as described (*Cash et al., 2009*). In brief, female mice (n = 5–10 per group) at 8–12 weeks of age were injected *i.p.* with zymosan A particles (250 mg/kg, 0.5 ml saline). Peritoneal lavage fluid was harvested with 5 ml PBS containing 1% FCS at 2 or 6 hr after the injection. Cells and residual zymosan particles were removed by centrifugation at 900×*g* for 4 min. Mice were grouped randomly, and studied without being blinded.

To detect apoptotic cells, peritoneal cells were suspended in PBS containing 2% FCS, incubated on ice with 1 mg/ml of biotinylated anti-Ly6B.2 (AbD Serotec, UK), followed by the incubation with 2 µg/ml PE-Cy7-labeled streptavidin (BD Pharmingen) and 5 µg/ml FITC-labeled anti-Ly6G (1A8; BD Pharmingen). After washing with Annexin V binding buffer, cells were stained with 2000-fold diluted Cy5-labeled AnnexinV (Biovision, Milpitas, CA) and 500 nM Sytox blue (Invitrogen) and analyzed by flow cytometry.

## Acknowledgements

We are grateful to Dr Jian-Fan Chen (Boston University School of Medicine) for *adenosine receptor A2a* knockout mice. We thank M Fujii for secretarial assistance.

## Additional information

### Funding

| Funder | Author |
| --- | --- |
| The Ministry of Education, Sports, and Culture in Japan | Shigekazu Nagata |
| Japan Science Technology | Shigekazu Nagata |

The funder had no role in study design, data collection and interpretation, or the decision to submit the work for publication.

### Author contributions

HY, Conception and design, Acquisition of data, Analysis and interpretation of data, Drafting or revising the article; TM, Acquisition of data, Drafting or revising the article; YU, Conception and design,

Analysis and interpretation of data; SN, Conception and design, Analysis and interpretation of data, Drafting or revising the article

## Ethics

Animal experimentation: All mice were housed in a specific pathogen-free facility at Kyoto University, Graduate School of Medicine. The committee of Kyoto University, Graduate School of Medicine, approved our study (Reference Number Med Kyo 12107).

## Addtional files

### Major dataset

The following previously published dataset was used:

| Author(s) | Year | Dataset title | Dataset ID and/or URL | Database, license, and accessibility information |
|---|---|---|---|---|
| Panchin Y, Kelmanson I, Matz M, Lukyanov K, Usman N, Lukyanov S | 2000 | A ubiquitous family of putative gap junction molecules | NM_019482.2; http://www.ncbi.nlm.nih.gov/nuccore/NM_019482.2 | Publicly available at the NCBI Nucleotide database (http://www.ncbi.nlm.nih.gov/nuccore). |

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
