## [Decision Letter]

Thank you for sending your work entitled “Immunosuppression via adenosine receptor activation by AMP released from apoptotic cells” for consideration at *eLife*. Your article has been favorably evaluated by a Senior editor and 2 reviewers, one of whom is a member of our Board of Reviewing Editors.

The following individuals responsible for the peer review of your submission have agreed to reveal their identity: David Wallach (Reviewing editor) and Martin Herrmann (peer reviewer).

The Reviewing editor and the other reviewer discussed their comments before we reached this decision, and the Reviewing editor has assembled the following comments to help you prepare a revised submission.

Both reviewers found your findings to be novel and important, and recommended publication. However, to place your new findings in perspective with regard to prior studies you are requested to address the following two questions in your manuscript, either by new data or just by discussing them:

1) Does the effect of extracellular adenosine fully account for the anti-inflammatory effect of apoptotic cells on macrophages that take them up, or only to a fraction of it? This can be answered by determining the actual concentrations of AMP and adenosine in the co-culture of apoptotic cells and macrophages and then comparing the extent of anti-inflammatory effect of the apoptotic cells (as assessed for example by quantifying the upregulation of one of the anti-inflammatory genes in the macrophages) to the anti-inflammatory effect of AMP and adenosine, when applied to the macrophages at those same concentrations, yet in the absence of apoptotic cells.

2) Is the extracellular adenosine in co-cultures of apoptotic cells and macrophages indeed fully derived from the apoptotic cells or, as was suggested previously by Köröskényi et al., in part also derived from the mononuclear phagocytes that take them up. An answer to the latter question can be obtained by comparing the concentrations of AMP+ Adenosine in cultures at which the mononuclear phagocytes are allowed to take up apoptotic cells to those in cultures of the same cells in which such uptake is prevented, for example by a semipermeable barrier.

---

## [Author Response]

*1) Does the effect of extracellular adenosine fully account for the anti-inflammatory effect of apoptotic cells on macrophages that take them up, or only to a fraction of it? This can be answered by determining the actual concentrations of AMP and adenosine in the co-culture of apoptotic cells and macrophages and then comparing the extent of anti-inflammatory effect of the apoptotic cells (as assessed for example by quantifying the upregulation of one of the anti-inflammatory genes in the macrophages) to the anti-inflammatory effect of AMP and adenosine, when applied to the macrophages at those same concentrations, yet in the absence of apoptotic cells*.

Cells expressing the caspase-resistant form of pannexin-1 do not release AMP during FasL-treated apoptosis (Figure 4). To address the above question, we treated the parental W3 and the W3 cell transformants expressing the caspase-resistant pannexin1, and co-cultured for 1 h with Bone-marrow derived macrophages (BMDM). The Thbs1 gene expression in BMDM was then quantified by real-time PCR, and was found that BMDM co-cultured with the FasL-treated wild-type W3 cells strongly express Thbs1. But, this was not observed when BMDM was co-cultured with FasL-treated W3 cells expressing the caspase-resistant form of Pannexin 1. The result was shown in Figure 4 of the revised manuscript and explained.

*2) Is the extracellular adenosine in co-cultures of apoptotic cells and macrophages indeed fully derived from the apoptotic cells or, as was suggested previously by Köröskényi et al., in part also derived from the mononuclear phagocytes that take them up. An answer to the latter question can be obtained by comparing the concentrations of AMP+ Adenosine in cultures at which the mononuclear phagocytes are allowed to take up apoptotic cells to those in cultures of the same cells in which such uptake is prevented, for example by a semipermeable barrier*.

To address the above question whether the macrophages engulfing apoptotic cells produce AMP or not, we first incubated mouse resident peritoneal macrophages (5 x 10^5^ cells) with FasL-treated mouse thymocytes (2 x 10^6^ cells) for 1 h for engulfment. The macrophages were thoroughly washed with medium to remove apoptotic cells, and cultured for 5 h. The AMP level increased in the supernatant was about 30 nM, corresponding to 30 pmoles per 5 x 10^5^ cells. This value is significantly lower than that produced by apoptotic cells (7 nmols per 10^7^ cells) (Figure 4). This point was briefly mentioned in the Discussion.